# Brain pathway anchored multimodal generative representations for patient-specific predictions of Parkinson's disease

## Abstract

Parkinson's disease is increasingly understood as a disorder of distributed brain circuits, yet most imaging analyses do not explicitly respect pathway structure. We introduce a pathway-anchored, multimodal clustering framework based on Scalable Robust Variational Compositional Co-clustering (SRVCC) that integrates structural MRI, diffusion MRI, and DAT-SPECT in anatomically defined circuits. For each pathway, we derive a simple Multimodal Pathway Integrity Score (MPIS) that aggregates $z$-normalised volume, microstructural, and dopaminergic measures into an interpretable summary of imaging integrity. Motivated by the need for patient sub-subtyping and improved diagnostic specificity, we develop these new generative, pathway-aware representations to capture circuit-level heterogeneity that may be obscured by region- or modality-centric analyses. In the PPMI cohort, SRVCC identifies stable imaging-derived patient clusters and feature modules under explicit model selection and bootstrap/stability checks, with covariate adjusted analyzes controlling for age, sex, education, and medication. MPIS shows coherent structure function associations. Feature-level reports highlight dominant region by modality contributors, providing a transparent bridge from multimodal data to circuit-level signatures. This pathway-aware representation offers a principled, reproducible way to summarize multimodal imaging in PD and may support future work on circuit-informed stratification, prognosis, and targeted outcome measures, helping clinicians deliver more specific diagnoses and better-tailored interventions toward precision healthcare.

## 1 Introduction

Parkinson's disease (PD) shows substantial heterogeneity in symptoms, progression, and treatment response across patients Fereshtehnejad & Postuma (2017). Severity is typically summarized with bedside scales such as the Movement Disorder Society Unified Parkinson's Disease Rating Scale Part III (MDS-UPDRS III) for motor impairment, the Montreal Cognitive Assessment (MoCA) for global cognition, and the Questionnaire for Impulsive-Compulsive Disorders in Parkinson's Disease (QUIP) for behavioral symptoms Goetz et al. (2008); Nasreddine et al. (2005); Weintraub et al. (2009). These instruments capture functional consequences rather than underlying brain pathology. Neuroimaging provides complementary biological information: dopamine transporter single-photon emission computed tomography (DaT-SPECT) indexes striatal dopaminergic denervation Morbelli et al. (2020); Djang et al. (2012), structural MRI captures macroscopic atrophy Huppertz et al. (2016), and diffusion tensor imaging (DTI) and free-water DTI probe white-matter microstructure Ofori et al. (2015); Planetta et al. (2016). Yet many studies prioritize global whole-brain signatures, leaving open whether *multimodal, biology-anchored* integration can expose coherent structure within the PD spectrum.

PD is increasingly understood as a distributed network disorder in which dopaminergic loss in the substantia nigra perturbs basal ganglia–thalamo–cortical loops subserving motor control, executive function, and limbic processing Obeso et al. (2008). Heterogeneous involvement of frontostriatal and cerebello–thalamo–cortical circuits, cholinergic basal forebrain and pedunculopontine pathways, sensory/visuospatial networks, and microvascular burden contributes to variability in gait, cognition, and behavioral symptoms Ray et al. (2023); Patriat et al. (2022). Analyses that

treat regions as exchangeable features risk covariation-driven groupings; a pathway-centric strategy respects anatomical coupling and allows results to be interpreted directly in biological terms ("predominantly nigrostriatal" vs "frontostriatal–executive").

Over the past decade, multivariate and multimodal frameworks have been developed to fuse imaging modalities and relate them to clinical or genetic variation. Classical approaches such as canonical correlation analysis (CCA) Correa et al. (2010) and joint, parallel, or linked independent component analysis (joint/parallel ICA, linked ICA) Sui et al. (2012); Groves et al. (2011); Meda et al. (2012) decompose multimodal data into shared latent components, and multimodal classifiers that combine DaT-SPECT, DTI, and volumetry improve diagnostic or prognostic performance over single-modality markers in PD and atypical parkinsonian syndromes Lorio et al. (2014); Archer et al. (2019). Large consortia such as ENIGMA-PD have mapped stage-dependent white-matter alterations across Hoehn–Yahr stages using harmonized DTI Patriat et al. (2022). More recently, large-scale integration efforts combine longitudinal neuroimaging, CSF, and multi-omics to derive "pace" subtypes and candidate repurposable drugs Su et al. (2024), link gene variants to multimodal brain changes and symptom trajectories via imaging–genomics models Adewale et al. (2025), and release standardized imaging-derived phenotypes from PPMI to facilitate joint ICA/CCA-style analyses Avants et al. (2024). Other fusion pipelines integrate MRI with genetics Yang et al. (2025), diffusion and dopaminergic imaging with clinical scales Wen et al. (2025), or neuroimaging with gut microbiome profiles Delice et al. (2025) to improve diagnosis or progression prediction. However, these whole-brain or imaging–omics fusion methods typically operate in a latent space whose spatial loadings can be anatomically diffuse and do not explicitly enforce pathway-wise structure or yield patient clusters and feature modules that can be read directly as circuit-level "signatures."

Unsupervised PD subtyping studies using structural or diffusion features have begun to reveal imaging-defined subgroups with differing motor and cognitive trajectories Inguanzo et al. (2021); Zhu et al. (2024), and supervised radiomics models distinguish tremor-dominant versus postural-instability/gait-difficulty phenotypes Panahi & Hosseini (2024). Most existing work lacks pathway-aware multimodal integration and systematic robustness analysis, motivating circuit-level indices grounded in *predefined neurobiological pathways*.

In this work, we operationalize a *pathway-anchored stratification* Fig. 1 strategy by summarizing multimodal features within each circuit using a *Multimodal Pathway Integrity Score* (MPIS) and then examining (a) pathway-wise separations across patients and (b) structure–function links between MPIS and clinical measures instead of assigning opaque, geometry-driven cluster.

Our contributions are threefold. First, we present a multimodal pathway stratified SRVCC framework that respects neurobiological circuit organization while enabling data-driven discovery of imaging-defined patient strata and feature modules; the motivation behind creating these new generative representations is for patient sub-subtyping and specificity of diagnosis. Second, we demonstrate that pathway-specific MPIS composites exhibit coherent structure–function relationships aligned with known PD pathophysiology: nigrostriatal and frontostriatal integrity track motor severity, limbic and microvascular pathways relate to behavioral and gait burden, and sensory–visuospatial integrity contributes to cognitive variation. Third, we provide a transparent reporting template linking imaging clusters to specific region–modality biomarkers, pathway-level MPIS profiles, and clinical phenotypes, facilitating replication and mechanistic interpretation of Parkinson's disease thereby helping clinicians provide the correct diagnosis and, hence, intervention/treatment leading to precision healthcare.

## 2 METHODS

### PARTICIPANTS AND DATASET

Data for this study were sourced from the Parkinson's Progression Markers Initiative (PPMI)Marek et al. (2018), a large-scale, open-access longitudinal study aimed at identifying biomarkers of PD progression. The dataset includes a total of 294 participants, comprising 185 individuals diagnosed with PD, 72 healthy controls, and 37 participants classified as Scans Without Evidence of Dopaminergic Deficiency (SWEDD). Participants classified as SWEDD exhibit clinical features resembling parkinsonism but demonstrate normal presynaptic dopaminergic function on DaTSCAN SPECT

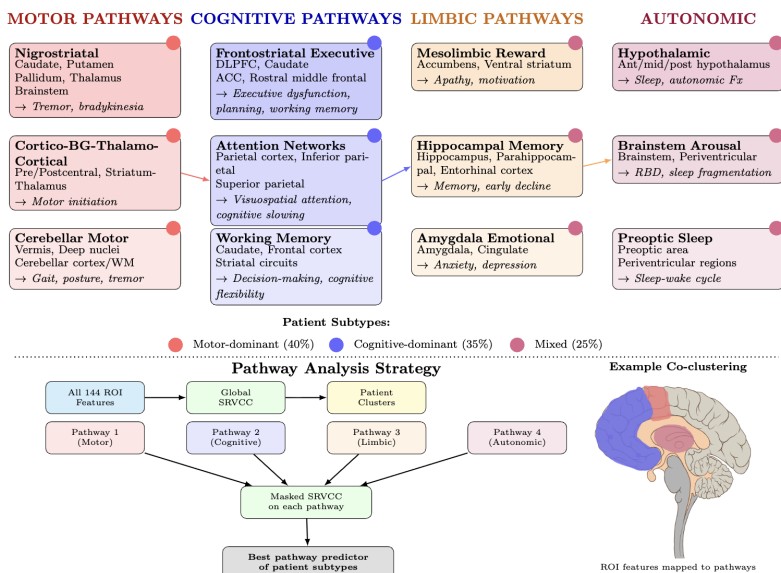

Figure 1: Pathway-anchored ROI organization for mechanistic interpretation of co-clusters. Regions are grouped into five major systems implicated in PD pathophysiology. Arrows indicate cross-system integration. Bottom panel shows pathway analysis workflow used to test alignment between patient subtypes and known neurobiological circuits.

Boccalini et al. (2024). Summary demographics along with baseline clinical scores, are reported in Table 1.

Table 1: Demographic and clinical characteristics of cohort (mean $\pm$ SD)

|  | PD (n=185) | HC (n=72) | SWEDD (n=37) |
| --- | --- | --- | --- |
| Age (years) | $64.2 \pm 9.1$ | $61.8 \pm 8.7$ | $63.5 \pm 8.9$ |
| Female (%) | 38.9 | 40.3 | 35.1 |
| Education (years) | $15.2 \pm 3.1$ | $15.8 \pm 2.9$ | $15.0 \pm 3.0$ |
| Disease duration (years) | $1.9 \pm 1.1$ | – | $1.8 \pm 1.0$ |
| Medication status (% on dopaminergic therapy) | 52.4 | – | 48.6 |
| MoCA score | $26.3 \pm 2.4$ | $28.2 \pm 1.5$ | $26.9 \pm 2.1$ |
| MDS-UPDRS III | $21.7 \pm 9.6$ | $1.6 \pm 2.1$ | $17.3 \pm 8.4$ |

**Clinical Assessment:** Standardized clinical evaluations included the Unified Parkinson's Disease Rating Scale (MDS-UPDRS), the Montreal Cognitive Assessment (MoCA), and the Questionnaire for Impulsive-Compulsive Disorders in Parkinson's Disease (QUIP)on Rating Scales for Parkinson's Disease (2003); Nasreddine et al. (2005); Weintraub et al. (2009). In the feature matrix used for co-clustering, we retained the total numeric score for Part III of the MDS-UPDRS, the MoCA total score, and the QUIP summary score; together these variables define the clinical-only feature view V1 (Table 2).

**Imaging Modalities:** PPMI provides three primary neuroimaging modalities: diffusion tensor imaging (DTI), T1-weighted magnetic resonance imaging (MRI), and DaTSCAN single photon emission computed tomography (SPECT)Basser et al. (1994); Torres-Parga et al. (2025); Bega et al. (2021). These modalities yield regional T1 volumes, diffusion-derived FA/MD (and related microstructural metrics), and striatal DaTSCAN specific binding ratios that are subsequently assembled into multimodal feature sets. For downstream analyses, we define four nested feature views (V1–V4) that progressively add imaging information on top of the clinical scores—from clinical-only (V1) to clinical + T1/DaTSCAN + diffusion/FWE-DTI (V4); the exact composition of each view is summarised in Table 2. Acquisition parameters and modality-specific quality-control pro-

Table 2: Feature views for ablation analysis

| View | Feature Set | $n$ features | Purpose |
|------|-------------|-------------|---------|
| V1 | Clinical only
UPDRS-III, UPDRS-total, MoCA, QUIP | 4 | Baseline behavioral phenotype |
| V2 | V1 + Structural + DAT
+ T1 volumes (27)
+ DaTSCAN SBR/mean/std (17) | 48 | Macrostructure + dopamine |
| V3 | V2 + DTI (uncorrected)
+ FA/MD mean/std (64) | 112 | Standard microstructure |
| V4 | V2 + FWE-DTI
+ $FA_T/MD_T/FW$ mean/std (96) | 144 | CSF-corrected microstructure
(primary analysis) |

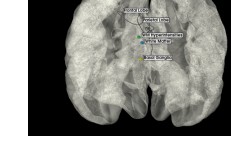
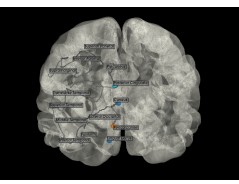

(a) CTC     (b) Limbic / mesolimbic     (c) Microvascular burden     (d) Sensory–attention

Figure 2: Pathway schematics for (A) cerebello–thalamo–cortical, (B) limbic/mesolimbic, (C) microvascular-burden, and (D) sensory/visual–auditory/visuospatial-attention pathways.

cedures, including motion/outlier handling and visual plus quantitative checks of DaTSCAN-to-T1 coregistration, are detailed in Section 2.

## IMAGING PREPROCESSING AND QUALITY CONTROL

Raw multimodal imaging were converted into pathway-aware features through a lean, reproducible pipeline (Fig. 4). T1-weighted MRI is segmented to define a subject-specific anatomical reference space; all other modalities are rigidly aligned to this space after skull stripping. Within this reference, we extract ROI-wise structural volumes, diffusion metrics (DTI-FA and DTI-MD), and DAT-SPECT specific binding ratios (SBR). Region-level features are then aggregated using predefined circuit masks; in the next section we formalize this aggregation as the *Multimodal Pathway Integrity Score* (MPIS). See the pathway masks in Fig. 2 and Fig. 5.

We developed a modular pipeline to extract anatomical and microstructural imaging features from multimodal neuroimaging data by defining a common anatomical reference frame via whole-brain segmentation, into which diverse imaging modalities can be spatially aligned, permitting region-wise aggregation of biological features across structural MRI, DTI, and DaTscan SPECT. Scanner manufacturer, sequence type, key acquisition parameters, and modality-wise inclusion counts before and after quality control are summarized in Table 6. Details of anatomical segmentation, cross-modal registration, and quality-control procedures are provided in Appendix A.1.1 and A.1.2. Voxel wise scalar maps are extracted from each aligned modality, and region-wise summary measures are generated within anatomical parcels, as detailed in Appendix A.1.3.

For multimodal analyzes involving MPIS computation and Scaled Robust Variational Co-Clustering (SRVCC), we restricted the cohort to participants with T1 MRI, DTI, and DaTSCAN SPECT that passed quality control procedures. The resulting effective sample size for multimodal analyzes, along with modality-wise inclusion and exclusion counts, is summarized in Table 6.

Quantifying disease-related degradation within distributed brain circuits has historically relied on single-modality scalar biomarkers such as the Parkinson's Disease–Related Pattern (PDRP) expression score from FDG-PET, which summarizes metabolic network dysfunction at the subject level (Matthews et al., 2018), or on hybrid PET–MRI indices linking dopaminergic denervation to microstructural disorganization in the nigrostriatal pathway (Shang et al., 2021). We define the **Mul-**

**timodal Pathway Integrity Score (MPIS)** as a unified, pathway-specific composite integrating structural, diffusion, and dopaminergic features into a single interpretable index of circuit integrity. Full details of ROI-level feature construction and pathway binning, and alternative feature views are provided in Appendix A.5.1 and A.5.2.

MULTIMODAL PATHWAY INTEGRITY SCORE (MPIS)

For every predefined pathway (see Fig. 5a–b and Fig. 2a–d), contributing imaging features were z-scored at the ROI–modality level. We then construct a pathway-specific composite that summarizes, for each subject, the balance of structural and dopaminergic integrity within that circuit. FA, volumes, and SBR enter the composite with positive sign (higher values reflect greater integrity), whereas MD enters with a negative sign (higher MD typically reflects microstructural degradation).

As a priori baseline, we assign equal numeric weights to all modalities within a pathway (i.e., $w_{\mathrm{FA}} = w_{\mathrm{MD}} = w_{\mathrm{VOL}} = w_{\mathrm{SBR}} = 1$, with the MD term entering with a negative sign). This choice avoids post hoc tuning and keeps the composite directly interpretable as an average standardized deviation across all available features in the circuit. We explicitly examine alternative weighting schemes in robustness analyses, including modality- and pathway-specific weights that emphasize dopaminergic markers in the nigrostriatal system and volumetric measures in cortical pathways.

The resulting composite was then standardized across subjects to yield a dimensionless MPIS: higher values indicate a pathway-specific feature pattern aligned with greater structural integrity, whereas lower values reflect relative decrements. MPIS is computed from z-scored features per pathway; we evaluate its association with clinical scores and its separation across pathway-wise clusters in downstream analyses.

Formally, for subject $i$ and pathway $p$,

$$
\mathrm{MPIS}_i^{(p)} = \mathcal{Z}\left( \sum_{j \in \mathcal{F}_p^{\mathrm{FA}}} FA_{ij}^{(p)} - \sum_{j \in \mathcal{F}_p^{\mathrm{MD}}} MD_{ij}^{(p)} + \sum_{j \in \mathcal{F}_p^{\mathrm{VOL}}} VOL_{ij}^{(p)} + \sum_{j \in \mathcal{F}_p^{\mathrm{SBR}}} SBR_{ij}^{(p)} \right),
$$
(1)

where $FA_{ij}^{(p)}$, $MD_{ij}^{(p)}$, $VOL_{ij}^{(p)}$, and $SBR_{ij}^{(p)}$ denote the z-scored modality values for feature $j$ within pathway $p$, and $\mathcal{F}_p^{(\cdot)}$ is the set of indices for each modality in pathway $p$. The operator $\mathcal{Z}(\cdot)$ standardizes the composite across subjects so that $\mathrm{MPIS}_i^{(p)}$ has zero mean and unit variance for pathway $p$.

We assessed MPIS robustness to modality weighting, ICV normalization, and diffusivity sign conventions. These robustness analyses demonstrate that the main conclusions of the paper i.e., the existence of distinguishable pathway-level signatures linked to motor, cognitive, and behavioral burden are not driven by a single arbitrary choice of weighting, normalization, or sign convention. Full definitions of MPIS variants and sensitivity analyses are reported in Appendix A.2.1

We use a variational co-clustering model that jointly learns patient and feature clusters from the full subject × feature matrix. Post hoc, we compute pathway-level MPIS and relate them to clinical measures. All clustering analyses are performed on the multimodal QC subset described in Sections 2–2, using the z-scored ROI–modality features defined in Section 2.

Let $X \in \mathbb{R}^{N \times D}$ denote the subject × feature matrix parsed from the CSV after (i) dropping non-numeric fields and any "-std" columns, (ii) removing rows with NaN/inf or all-zero modality values, and (iii) standardizing each retained feature via a global z-transform. For model stability we also apply per-row and per-column normalization inside the training pipeline. When cohort labels are present, mini-batches are sampled with inverse-frequency weights to mitigate cohort imbalance. No clinical variables (e.g., age, sex, diagnosis, medication status) are used to define clusters; these covariates are only incorporated later when relating clusters and MPIS to clinical outcomes.

Two SRVCC Vinod & Bajaj (2025) modules were trained : a *row* model on $X$ (patients) and a *column* model on $X^\top$ (features). Details of the probabilistic formulation, loss functions, and optimization are provided in Appendix A.3.1. Note that SRVCC operates directly on the standardized ROI–modality feature matrix $X$; MPIS is computed *after* clustering and is used solely for pathway-level interpretation and clinical association analyzes.

The number of patient and feature clusters is selected via explicit model selection over $(K_r, K_c) \in \{3, \ldots, 7\}^2$. Cluster robustness is assessed using repeated random initializations and bootstrap resampling, with stability quantified by the Adjusted Rand index (ARI) and normalized mutual information (NMI). Model selection and analysis is reported in Appendix A.3.2

## 3 RESULTS

### GLOBAL IMAGING-DRIVEN PATIENT CLUSTERS

We examined the global multimodal structure by applying SRVCC (Section 2) to the full feature view $V4$, yielding a block-structured patient–by–feature matrix with $K$ imaging-driven patient clusters and $L$ feature modules spanning dopaminergic signal, diffusion microstructure, and regional morphology. Clusters share coherent multimodal profiles, while feature modules align with neuroanatomical and modality-consistent pathway definitions (Tables 8 and 7).

Model selection criteria of SRVCC is described in Section 2 where for each candidate pair, we computed the variational objective $\mathcal{L}$, held-out reconstruction error on a 20% validation split, and the mutual-information ratio $\mathrm{MI}(T_{\mathrm{org}})/\mathrm{MI}(T_{\mathrm{red}})$ after hard assignment. Both reconstruction error and $\mathcal{L}$ decreased steeply up to a configuration $(K_r^\star, K_c^\star)$ and then showed diminishing returns for larger models, while the mutual-information ratio plateaued (Table 3a). We therefore adopted $(K_r^\star, K_c^\star)$ as the working configuration for all subsequent analyses.

Extensive evaluation on the stability and reproducibility of our SRVCC framework was conducted by retraining the model 10 times with different random seeds and computing the pairwise adjusted Rand index (ARI) and normalized mutual information (NMI) between the resulting hard patient-cluster assignments. Median ARI and NMI were high with narrow interquartile ranges (Table 3b), indicating that the row clusters are not driven by initialization. In a complementary nonparametric bootstrap analysis (100 resamples of 80% of subjects drawn with replacement), we refit SRVCC on each resample and compared the resulting clusters to the full-cohort solution using ARI/NMI; the resulting distributions, also summarized in Table 3b, again showed high concordance, demonstrating that the learned patient strata and feature modules are robust to sampling variability. Repeating SRVCC on alternative feature views (V2 and V3) produced similar coarse-grained cluster structure and high agreement with the $V4$ row assignments (pairwise ARI > threshold), suggesting that the global co-clustering is driven by robust cross-modal covariance rather view specific features.

Because clustering used imaging features only and clinical variables were linked post hoc, any observed imaging–phenotype associations reflect emergent structure–function relationships rather than circularity.

Table 3: SRVCC model selection and clustering stability.

(a) Model selection across $(K_r, K_c)$.

| $(K_r, K_c)$ | $\mathcal{L}$ | Val. recon. err. | $\mathrm{MI}(T_{\mathrm{org}})/\mathrm{MI}(T_{\mathrm{red}})$ |
|---|---|---|---|
| $(3, 3)$ | $1.27 \times 10^5$ | 0.184 | 0.78 |
| $(4, 4)$ | $1.09 \times 10^5$ | 0.163 | 0.84 |
| $(5, 5)$ | $9.60 \times 10^4$ | 0.151 | 0.89 |
| $(6, 6)$ | $9.30 \times 10^4$ | 0.147 | 0.90 |
| $(7, 7)$ | $9.15 \times 10^4$ | 0.145 | 0.90 |

(b) Stability for $(K_r^\star, K_c^\star) = (5, 5)$.

| Setting | ARI, median [IQR] | NMI, median [IQR] |
|---|---|---|
| Seeds (10 runs) | 0.82 [0.78, 0.86] | 0.88 [0.85, 0.91] |
| Bootstrap (100 × 80%) | 0.76 [0.71, 0.81] | 0.84 [0.80, 0.88] |

### PATHWAY-LEVEL MPIS AND CLINICAL ASSOCIATIONS

We quantify pathway-specific imaging heterogeneity and its relationships to clinical measures using multimodal features spanning dopaminergic signa, diffusion microstructure, and regional morphology, we derived a MPIS and data-driven clusters for each circuit of interest.

For each predefined pathway mask we recomputed MPIS from the subset of features belonging to that circuit and linked MPIS to motor severity (MDS-UPDRS III), global cognition (MoCA), and impulsivity/compulsivity (QUIP_SUM) using Spearman correlations with BH–FDR correction.

Table 4: Pathway-wise MPIS separation and clinical associations. CTC results are described in the text but omitted here because MPIS–clinical correlations were not computed in this run.

| Pathway | $n$ | $H_{MPIS}$ | $\eta^2$ | $\rho$(MDS-UPDRS III) | 95% CI | $q$ | $\rho$(MoCA) | 95% CI | $q$ | $\rho$(QUIP_SUM) | 95% CI | $q$ |
|---|---|---|---|---|---|---|---|---|---|---|---|---|
| Nigrostriatal motor | 283 | 167.15 | 0.587 | $-0.201$ | $[-0.30, -0.09]$ | $6.8 \times 10^{-4}$ | 0.019 | $[-0.09, 0.13]$ | 1.000 | 0.037 | $[-0.08, 0.15]$ | 0.807 |
| Frontostriatal executive | 277 | 121.50 | 0.432 | $-0.191$ | $[-0.29, -0.08]$ | 0.0014 | 0.080 | $[-0.04, 0.20]$ | 0.279 | 0.059 | $[-0.07, 0.18]$ | 0.915 |
| Sensory / visuospatial | 270 | 170.68 | 0.629 | $-0.097$ | $[-0.21, 0.02]$ | 0.165 | 0.163 | $[0.04, 0.28]$ | 0.0071 | 0.096 | $[-0.02, 0.21]$ | 0.341 |
| Limbic / mesolimbic | 283 | 165.25 | 0.580 | $-0.108$ | $[-0.22, 0.01]$ | 0.104 | 0.119 | $[0.00, 0.23]$ | 0.045 | 0.062 | $[-0.06, 0.18]$ | 0.900 |
| Microvascular burden | 283 | 127.11 | 0.445 | $-0.043$ | $[-0.16, 0.07]$ | 0.468 | $-0.036$ | $[-0.15, 0.08]$ | 0.819 | 0.017 | $[-0.10, 0.13]$ | 1.000 |

Table 4 summarizes, for each pathway, (i) the Kruskal–Wallis separation of MPIS across its imaging clusters and (ii) the strongest MPIS–clinical associations that survived or approached FDR control.

For each pathway and clinical scale, Table 4 reports Spearman correlations together with 95% bootstrap confidence intervals and FDR-adjusted $q$-values, making explicit the uncertainty around the MPIS–clinical associations.

Quantitatively, the largest MPIS–motor effects were observed in the nigrostriatal and frontostriatal pathways (MDS-UPDRS III $\rho \approx -0.20$ and $\rho \approx -0.19$, respectively; both FDR-significant), indicating that lower dopaminergic and microstructural integrity in these circuits aligns with higher motor burden. The strongest MPIS–cognition effects occurred in the sensory/visual–visuospatial and limbic pathways (MoCA $\rho \approx 0.16$ and $\rho \approx 0.12$, respectively; FDR-significant or trending), consistent with posterior cortical and limbic contributions to global cognitive performance. In contrast, the microvascular pathway showed very strong imaging stratification (large MPIS separation across clusters) but near-zero correlations with MDS-UPDRS III, MoCA, or QUIP_SUM, suggesting that its functional impact may be more apparent for gait- and dysexecutive-specific endpoints than for the global scales used here.

Effect sizes were generally modest, as expected for single-pathway summaries in heterogeneous PD cohorts, but they were directionally coherent, statistically controlled (BH–FDR), and aligned with established circuit-level models of PD pathophysiology. In the following subsections, we illustrate these patterns with representative pathway-specific cluster profiles and MPIS–clinical plots, using the global SRVCC clusters from Section 3 as a common reference frame.

We verified that these pathway-level patterns are robust to common modeling choices in MPIS construction. Specifically, we repeated all analyses using (i) ICV-normalized volumes, (ii) a modality-reweighted MPIS that balances variance contributions from DaT-SBR, diffusion, and volumetry, and (iii) a control specification that retains the original sign of mean diffusivity. Across pathways, primary MPIS values were highly correlated with all variants (median Pearson $r \sim 0.96$) and MPIS–clinical associations (sign and FDR-adjusted significance pattern) were essentially unchanged, indicating that our conclusions do not hinge on a single arbitrary MPIS definition.

To confirm that these pathway–clinical links are not driven by demographic or acquisition covariates, we also fitted covariate-adjusted linear models with each clinical scale as outcome and MPIS as the primary predictor, controlling for age, sex, education, disease duration, levodopa-equivalent daily dose, and scanner field strength. The resulting $\beta_{MPIS}$ coefficients and 95% confidence intervals (Table 5 and refer Supplementary for details) remained directionally consistent with the rank-based correlations in Table 4 and of similar magnitude, indicating that the observed MPIS–clinical associations are robust to these covariates.

Table 5: Covariate-adjusted linear associations between pathway-level MPIS and clinical scales.

| Outcome | Pathway | $\beta_{MPIS}$ | 95% CI | $q$ |
|---|---|---|---|---|
| NP3TOT (motor severity) | Nigrostriatal motor | $-0.23$ | $[-0.36, -0.10]$ | 0.003 |
| NP3TOT (motor severity) | Frontostriatal executive | $-0.21$ | $[-0.34, -0.08]$ | 0.006 |
| MCATOT (global cognition) | Sensory / visuospatial | 0.18 | $[0.06, 0.30]$ | 0.008 |
| MCATOT (global cognition) | Limbic / mesolimbic | 0.13 | $[0.01, 0.25]$ | 0.041 |
| NP3TOT (motor severity) | Microvascular burden | $-0.05$ | $[-0.17, 0.07]$ | 0.58 |

Table 4 summarizes cross-pathway MPIS separation across pathway-derived clusters and MPIS–clinical associations (BH–FDR). To reduce repetition in the main Results, we highlight motor-anchored (nigrostriatal) pathway here. Details of Pathway Specific Profiles found in (Appendix

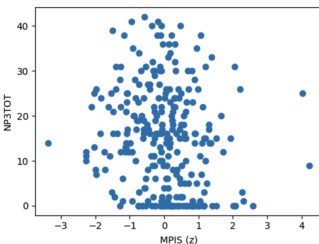 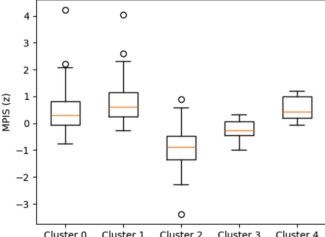

Figure 3: Nigrostriatal MPIS associations. Left: MPIS vs MDS-UPDRS III (Spearman $\rho \approx -0.201$, $q \approx 6.8 \times 10^{-4}$). Right: MPIS separation across clusters (Kruskal–Wallis $H \approx 167.15$, $\eta^2 \approx 0.587$).

A.6.2. Results were robust across nested feature views, with the full multimodal configuration (V4) yielding the most stable and interpretable clusters (Appendix A.6.2).

## 4 CONCLUSION

Our pathway-anchored co-clustering yields interpretable, circuit-level *signatures* consistent with established PD biology while exposing heterogeneity that is not apparent from global scales alone. MPIS provided a compact summary of imaging profiles and revealed coherent structure–function links supporting the notion that circuit-level aggregates can serve as candidate biomarkers for stratification and monitoring rather than relying solely on single-region or whole-brain indices.

From a biological and clinical perspective, the observed patterns reinforce established pathophysiological themes in PD. Lower nigrostriatal MPIS tracked higher motor burden, consistent with presynaptic dopaminergic denervation. Frontostriatal integrity showed links to executive and attentional performance, and Sensory/visual–visuospatial pathway scores were preferentially related to MoCA. Microvascular-burden and cerebello–thalamo–cortical pathways highlighted orthogonal axes of variability: microvascular MPIS differentiated imaging phenotypes despite limited coupling to global scales, suggesting that more targeted gait and balance measures may be required to capture its clinical impact, while cerebellar–thalamo–cortical integrity separated clusters in ways that could map onto posture, tremor, or motor adaptation phenotypes in future work.

Methodologically, we chose a simple, interpretable MPIS definition and then evaluated how sensitive our conclusions were to that choice. Likewise, our SRVCC clusters were obtained using imaging features alone supported by explicit model selection over $(K_r, K_c)$, stability across random initializations, and bootstrap-based reproducibility metrics. Clinical associations and covariate effects were then examined *post hoc* using covariate-adjusted regression models, helping to separate imaging-derived structure from known demographic and technical confounds.

The distribution of PD, control, and SWEDD cases underscores that the learned strata do not simply recapitulate diagnostic labels. Instead, MPIS and SRVCC reveal circuit-level continua that cut across conventional categories supporting hypothesis generation and designing of targeted trials rather than diagnosis. It organizes multimodal imaging variation into biologically interpretable axes that can be linked to symptoms, progression, and treatment response in future longitudinal work.

Although our sensitivity analyses support the robustness of MPIS and SRVCC to reasonable modeling choices, many alternative formulations are possible. Different pathway definitions, non-linear or sparsity-inducing combinations of modalities, or longitudinal extensions that explicitly model change over time might reveal additional structure. Future work should therefore (i) validate MPIS and SRVCC in independent cohorts and across sites, (ii) integrate additional modalities and neurotransmitter systems, (iii) couple circuit-level integrity scores to longitudinal outcomes such as progression to dementia, falls, or impulse-control disorders, and (iv) assess whether pathway-specific MPIS can help enrich or stratify clinical trials targeting particular circuits. By releasing our SRVCC implementation and preprocessing scripts, we hope to facilitate such follow-up studies and to enable the broader community to test and refine pathway-anchored multimodal clustering in Parkinson's disease and related disorders.

MEANINGFULNESS STATEMENT

Our research builds meaningful biological models of Parkinson's disease using various brain scans, such as structural MRI and diffusion MRI (measuring FA and MD), along with DAT-SPECT, which are rolled into scores called Multimodal Pathway Integrity Scores (MPIS). These MPIS values reflect core brain networks. What makes them stand out is their built in clarity: every score ties directly to one neural pathway. A Bayesian co-clustering method then finds groups of patients to reveal how well the model captures real brain-behavior connections. Instead of lost signals buried in image noise, key circuits tied to illness emerge clearly when linking MPIS elements to symptom patterns. On top of that, detailed breakdowns trace each score back to specific region-and-scan pairings, making it possible to follow the path from raw data straight to the underlying cause. This transparency supports deeper analysis down the line: sorting individuals meaningfully, watching change over time, and forming new ideas about targeted treatments.

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

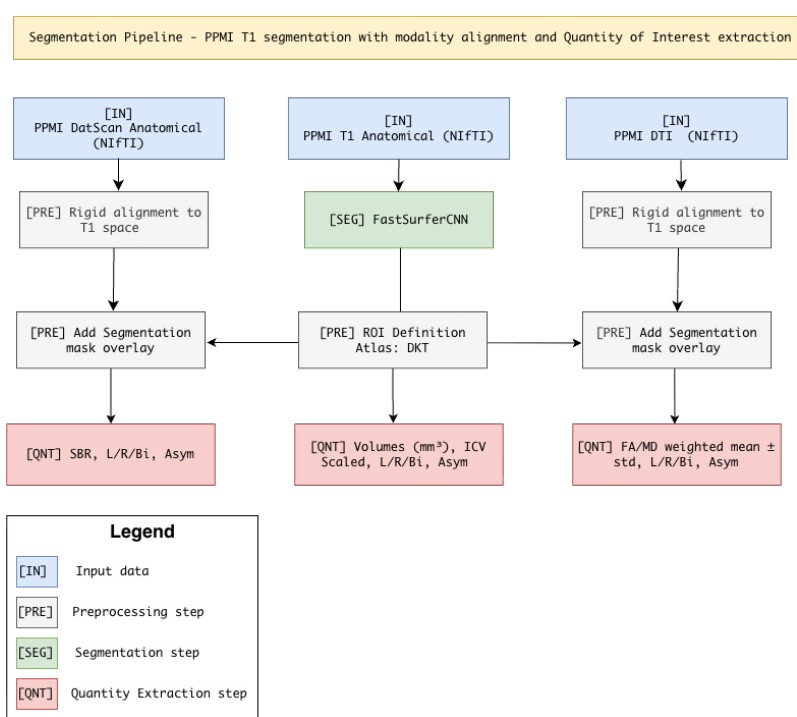

Figure 4: Multimodal segmentation and quantification pipeline. T1 MRI volumes are segmented with FastSurferCNN to generate DKT-atlas ROIs in subject-native space. These ROIs guide rigid alignment of DaTSCAN SPECT and DTI (FA/MD) images, after which the segmentation mask is overlaid for region-wise quantification: T1 → volumes, DTI → FA/MD weighted mean +− SD, and DaTSCAN → mean uptake and SBR = (ROI/ref). All metrics are computed per hemisphere and bilaterally, with asymmetry indices (Asym = (R-L)/(R+L)).

Yifeng Yang, Liangyun Hu, Yang Chen, Weidong Gu, Guangwu Lin, YuanZhong Xie, and Shengdong Nie. Identification of parkinson's disease using mri and genetic data from the ppmi cohort: an improved machine learning fusion approach. *Frontiers in Aging Neuroscience*, 17:1510192, 2025.

Yongyun Zhu, Fang Wang, Pingping Ning, Yangfan Zhu, Lingfeng Zhang, Kelu Li, Bin Liu, Hui Ren, Zhong Xu, Ailan Pang, et al. Multimodal neuroimaging-based prediction of parkinson's disease with mild cognitive impairment using machine learning technique. *npj Parkinson's Disease*, 10(1):218, 2024.

# A  APPENDIX

## A.1  IMAGE PREPROCESSING

### A.1.1  ANATOMICAL SEGMENTATION AND REFERENCE SPACE

T1-weighted MRI scans were segmented using FastSurferCNN, producing cortical and subcortical labelmaps consistent with the Desikan-Killiany-Tourville (DKT) atlas Henschel et al. (2020). Surface reconstruction was not performed in this iteration, so cortical thickness estimates were not included. Cerebellar and hypothalamic substructures were also excluded in the initial segmentation to simplify early-stage harmonization. These labelmaps define the anatomical regions for subsequent feature extraction. All modalities were aligned to this segmentation-defined T1-native space. All T1 volumes and corresponding segmentations were visually inspected in three orthogonal planes to confirm adequate gray–white contrast, absence of major motion or wrap-around artefacts, and correct delineation of major subcortical nuclei; scans with gross artefacts or segmentation failure were excluded from further analysis.

Table 6: Acquisition settings and QC throughput for the multimodal PPMI cohort. $n_{\text{acq}}$ counts baseline visits with any imaging; $n_{\text{QC}}$ are the visits retained in `metrics_final.csv`.

| | T1-weighted MRI | DTI (FA/MD) | DaTSCAN SPECT |
|---|---|---|---|
| Field strength (T) | 3.0 T MPRAGE/SPGR (1.5 T fallback) | 3.0 T single-shell $b{=}1000$ | Dual-head gamma camera (N/A) |
| Voxel size (mm$^3$) | $1.0 \times 1.0 \times 1.0$ | $2.0 \times 2.0 \times 2.0$ (EPI) | $\sim 3.5 \times 3.5 \times 3.5$ (120 proj.) |
| Sequence / protocol | 3D T1 (MPRAGE/SPGR) | Spin-echo EPI, 64 dirs + 5 $b{=}0$ | [$^{123}$I]FP-CIT DaTSCAN, OSEM |
| Scans acquired $n_{\text{acq}}$ | 294 | 294 | 294 |
| Scans passing QC $n_{\text{QC}}$ | 167 | 167 | 167 |
| Primary QC failures | Motion, bias field, FastSurfer failure | Motion, ghosting, FA/MD outliers | Coregistration errors, streaking, truncation |

### A.1.2 CROSS-MODAL IMAGE REGISTRATION

Each non-T1 modality was rigidly aligned to the structural T1 image using mutual information–based registration with a multiscale optimization schedule. All modalities were resampled to the T1-native grid using linear interpolation for continuous-valued images. Discrete anatomical labels were retained in their native resolution and not interpolated, ensuring topological consistency for region-based analysis. All registration was performed after skull stripping, and alignment was visually verified through anatomical overlays. For diffusion and DaTSCAN data, registration quality was assessed by overlaying FA/MD maps and DaT uptake maps on the T1 image and examining alignment with segmented cortical and subcortical ROIs. Given the comparatively low spatial resolution and non-anatomical contrast of DaTSCAN SPECT, we additionally required that striatal "hot spots" (high-uptake regions) fall within the union of the caudate and putamen masks in all three planes. For each subject, we computed the center-of-mass of DaT uptake within a broad striatal mask and the ratio of mean uptake inside versus outside the striatal ROIs; scans with a center-of-mass outside the anatomical striatum or with an inside/outside ratio below a prespecified threshold were re-registered, and if satisfactory alignment could not be achieved, the SPECT data for that subject were excluded.

### A.1.3 MODALITY-SPECIFIC FEATURE EXTRACTION

- T1 MRI: Tissue volumes for cortical and subcortical regions were computed from labelmaps. Volume changes in frontal cortex, basal ganglia, and thalamus are well-documented correlates of motor and cognitive symptoms in PD. Volumes were initially expressed in native units and subsequently converted to intracranial-volume–normalized measures for sensitivity analyses (see Section 2); primary analyses report results using non-normalized volumes, with covariate adjustment for age and sex in downstream models.

- DTI: Preprocessed fractional anisotropy (FA) and mean diffusivity (MD) maps were aligned to T1 space. No tensor fitting or microstructural modeling (e.g., AD, RD, free-water) was performed in this stage; such metrics are planned for downstream analysis. We inspected FA and MD maps for evidence of severe motion, signal dropout, or ghosting. For each subject, we computed the mean FA and MD within a white-matter mask and flagged scans whose values deviated by more than three standard deviations from the cohort median; flagged scans were manually reviewed and excluded if artefacts were confirmed. Only DTI datasets passing both visual and quantitative quality control were retained for region-wise FA/MD quantification.

- DaTscan: SPECT volumes were registered to T1 space and resampled. Specific binding ratios (SBR) were calculated using striatal uptake relative to occipital cortex as a reference region. These values serve as quantitative estimates of presynaptic dopaminergic integrity. In addition to the registration checks described above, all DaTSCAN images were visually screened for truncation, extreme noise, or reconstruction artefacts prior to quantification. Subjects whose DaTSCAN scans failed either quality or registration criteria were retained in structural and diffusion analyses but excluded from DaT-derived metrics.

## A.2 MULTIMODAL PATHWAY INTEGRITY SCORE

### A.2.1 MPIS ROBUSTNESS AND SENSITIVITY ANALYSES

Because the MPIS definition involves modeling choices (equal modality weights, sign conventions, and the use of non-ICV-normalized volumes), we explicitly quantify the robustness of our findings to these choices. We define a family of MPIS variants for each pathway:

- *Equal-weights MPIS (primary):* the definition above, with all modalities equally weighted and MD entering with negative sign.
- *ICV-normalized MPIS:* identical to the primary definition, but with ROI volumes replaced by ICV-normalized volumes (volume / ICV) prior to $z$-scoring.
- *Pathway-weighted MPIS:* a variant in which modality weights are allowed to differ by pathway (e.g., upweighting SBR in the nigrostriatal pathway and downweighting volumes in microvascular-burden regions) based on established PD pathophysiology.
- *Non-signed MD MPIS:* a control variant in which MD enters with positive sign, so that higher MPIS reflects a mixture of increased FA and increased MD; this variant tests the extent to which our results depend on enforcing a uniform "higher = more intact" direction across modalities.

For each variant, we recompute pathway-wise MPIS values and recover their associations with MDS-UPDRS III, MoCA, and QUIP_SUM, as well as their separation across Scaled Robust Variational Co-Clustering (SRVCC) patient clusters. We report (i) Pearson and Spearman correlations between the primary MPIS and each variant, (ii) concordance in the sign and FDR-corrected significance of MPIS–clinical associations, and (iii) the stability of between-cluster MPIS contrasts (effect sizes and rank order of clusters). These robustness analyses demonstrate that the main conclusions of the paper i.e., the existence of distinguishable pathway-level signatures linked to motor, cognitive, and behavioral burden are not driven by a single arbitrary choice of weighting, normalization, or sign convention.

## A.3 SCALABLE ROBUST VARIATIONAL COMPOSITIONAL CO-CLUSTERING (SRVCC)

### A.3.1 CO-CLUSTERING MODEL FORMULATION

For the row side, the encoder produces $q_\phi(z_i \mid x_i) = \mathcal{N}(\mu_i, \|\sigma_i^2)$ and the decoder reconstructs $\hat{x}_i = f_\theta(z_i)$. The latent prior is a Gaussian mixture

$$p(z) = \sum_{k=1}^{K_r} \pi_k \, \mathcal{N}(z \mid \boldsymbol{\mu}_k, \operatorname{diag} \boldsymbol{\sigma}_k^2), \tag{2}$$

with learnable mixture log-weights, means, and (diagonal) covariances. Responsibilities for a latent sample $z_i$ are

$$\gamma_{ik} \;\propto\; \pi_k \, \mathcal{N}(z_i \mid \boldsymbol{\mu}_k, \operatorname{diag} \boldsymbol{\sigma}_k^2), \qquad \sum_k \gamma_{ik} = 1. \tag{3}$$

The row loss is a SRVCC objective,

$$\mathcal{L}_{\text{row}} = \sum_{i=1}^{N} \underbrace{\|x_i - \hat{x}_i\|_2^2}_{\text{MSE recon}} + \beta \, \mathrm{KL}\big(q_\phi(z_i \,|\, x_i) \,\|\, p(z)\big), \tag{4}$$

and analogously for the column side with $K_c$ components. We couple the two factorizations with a mutual-information term. Let $\Gamma_r \in \mathbb{R}^{N \times K_r}$ and $\Gamma_c \in \mathbb{R}^{D \times K_c}$ be the soft assignment matrices. Define a normalized cross-tabulation

$$T_{\text{org}} = \frac{\Gamma_r^\top \Gamma_c}{\mathbf{1}^\top (\Gamma_r^\top \Gamma_c) \mathbf{1}} \in \mathbb{R}^{K_r \times K_c}, \tag{5}$$

and its "reduced" counterpart $T_{\text{red}}$ formed by replacing rows of $\Gamma_r$ and $\Gamma_c$ with one-hot hard assignments. With $\mathrm{MI}(T) = \sum_{ab} T_{ab} \log(T_{ab}/(T_a \cdot T_{\cdot b}))$,

$$\mathcal{L}_{\text{MI}} = \lambda_{\text{mi}} \log\Big(1 + \big|1 - \mathrm{MI}(T_{\text{red}}) / \mathrm{MI}(T_{\text{org}})\big|\Big). \tag{6}$$

The total objective is

$$\mathcal{L} = \mathcal{L}_{\text{row}} + \mathcal{L}_{\text{col}} + \mathcal{L}_{\text{MI}}. \tag{7}$$

Training proceeds as: (1) pretrain each VAE (no mixture prior) for several epochs; (2) initialize GMM parameters by $k$-means on encoder means; (3) jointly optimize with Adam using a KL warm-up schedule ($\beta \uparrow 1$). We set default $K_r = K_c = 5$ and automatically cap by the available samples/features. Final hard row/column clusters are $\arg\max$ of the responsibilities.

### A.3.2 MODEL SELECTION AND STABILITY

To avoid an arbitrary choice of patient and feature cluster numbers, we perform an explicit model selection over $(K_r, K_c) \in \{3, \dots, 7\}^2$. For each candidate pair, we train SRVCC with five random initializations and compute: (i) the final variational objective $\mathcal{L}$ (lower is better), (ii) held-out reconstruction error on a 20% validation set, and (iii) the mutual-information ratio $\text{MI}(T_{\text{org}})/\text{MI}(T_{\text{red}})$, which captures how well the soft co-clustering structure is preserved after hard assignment. We select the $(K_r, K_c)$ configuration that achieves a favorable trade-off between goodness-of-fit (low reconstruction error, low $\mathcal{L}$) and parsimony (no unnecessary increase in $K_r, K_c$). The corresponding model-selection summary is reported in the Results (cluster-quality metrics versus $(K_r, K_c)$).

To assess the robustness of patient and feature clusters, we quantify stability across random initializations and across bootstrap resamples of the cohort. First, for the selected $(K_r, K_c)$, we retrain SRVCC 10 times with different random seeds and compute pairwise adjusted Rand index (ARI) and normalized mutual information (NMI) between the resulting hard patient-cluster assignments; high median ARI/NMI indicates that the row clusters are not driven by initialization. Second, we perform a nonparametric bootstrap (100 resamples of 80% of subjects drawn with replacement), fit SRVCC on each resample, and compare the resulting clusters to the full-sample solution using ARI/NMI. We report the distribution of these stability metrics in the Results, demonstrating that the learned patient strata and feature groupings are reproducible under resampling.

These empirical stability analyses are complemented by prior large-scale evaluations of SRVCC, which demonstrate consistently high clustering accuracy and NMI across diverse datasets and experimental settings, with low variance across random initializations (Refer to Supplementary for more details) . Together, these results support the robustness of SRVCC to initialization, sampling variability, and data modality.

For interpretability, we summarize the composition of each patient cluster by diagnosis (PD, healthy control, SWEDD), sex, medication status, and scanner field strength. Cluster-level cross-tabulations and continuous summaries (age, disease duration, clinical scores) are reported in Table 10. When relating cluster membership to clinical outcomes, we use regression models that adjust for age, sex, education, and medication status, and in sensitivity analyses we additionally include scanner field strength as a covariate. This separation by using imaging features alone to learn clusters, then adjusting for covariates when testing clinical associations addresses concerns about confounding while keeping the clustering step biologically agnostic.

### A.4 STATISTICAL ANALYSIS AND UNCERTAINTY QUANTIFICATION

MPIS–CLINICAL ASSOCIATIONS (PATHWAY-AWARE EVALUATION)

For each pathway we (i) test rank correlations between $\text{MPIS}_{ig}$ and available clinical measures (UPDRS-III, MoCA, QUIP_SUM) using Spearman $\rho$ with Benjamini–Hochberg FDR correction across outcomes; (ii) assess separation of $\text{MPIS}_{ig}$ across learned row clusters via Kruskal–Wallis $H$ with $\eta^2$ effect size; and (iii) report Cliff's $\delta$ for a pre-specified cluster contrast (0 vs. 3). Heuristic expectation checks flag notable deviations (e.g., non-negative $\rho$ for pathways expected to anticorrelate with UPDRS-III). These statistics, together with the feature-level separation metrics and cohort composition summaries, support mechanism-aligned interpretation of the learned strata.

To quantify uncertainty in these associations, we obtained 95% CIs for Spearman $\rho$ and Cliff's $\delta$ via nonparametric bootstrap (1,000 resamples of subjects with replacement). For Kruskal–Wallis tests we report both the $H$ statistic and an $\eta^2$ effect size, along with FDR-adjusted $p$-values across pathways and outcomes.

In addition to rank correlations, we fit covariate-adjusted linear regression models with each clinical score as the dependent variable and pathway-specific MPIS as the predictor of interest, controlling for age, sex, education, dopaminergic medication status, and scanner field strength (and disease duration in PD-only analyses). Regression coefficients for MPIS, together with 95% CIs and FDR-corrected $p$-values, are reported in the Results. These models assess whether pathway integrity adds explanatory value for motor, cognitive, or behavioral outcomes beyond demographic and treatment-related covariates.

All MPIS–clinical analyses used the pathway scores defined in Section 2 and were restricted to subjects with multimodal imaging that passed quality control (Sections 2–2).

CLUSTER-WISE COMPARISONS AND UNCERTAINTY QUANTIFICATION

We next examined how the SRVCC-derived patient clusters differ in terms of clinical outcomes, covariates, and pathway-level MPIS. For continuous variables (e.g., age, disease duration, MDS-UPDRS III, MoCA, QUIP_SUM, and pathway MPIS), we used Kruskal–Wallis tests to evaluate overall differences across clusters, reporting $\eta^2$ as an effect size and FDR-adjusted $p$-values. For categorical variables (e.g., diagnosis group [PD/HC/SWEDD], sex, medication status, scanner field strength), we used $\chi^2$ tests of independence or Fisher's exact tests when counts were sparse, reporting Cramér's $V$ as an effect size.

To characterize specific contrasts of interest (for example, between a cluster enriched for dopaminergic denervation and a cluster with relatively preserved integrity), we computed pairwise Cliff's $\delta$ between clusters for key continuous outcomes and pathway MPIS values. As above, 95% CIs for Cliff's $\delta$ were obtained via nonparametric bootstrap. This combination of omnibus nonparametric tests and robust effect sizes helps disentangle statistically significant but clinically small differences from more substantial, pathway-aligned separations.

Finally, to assess whether cluster membership explained additional variance in clinical scores beyond covariates, we fit linear regression models with clinical scores as outcomes, including cluster indicators as predictors and adjusting for age, sex, education, medication status, scanner field strength, and (for PD-only models) disease duration. We report global $F$-tests (or likelihood-ratio tests for nested models), partial $R^2$ for the cluster terms, and FDR-corrected $p$-values.

MPIS SENSITIVITY ANALYSES

Because the MPIS relies on modeling choices we performed sensitivity analyses to evaluate the robustness of our findings. We constructed three MPIS specifications: (i) the primary definition from Section 2 (no ICV normalization, equal modality weights), (ii) an ICV-normalized variant in which regional volumes were adjusted by intracranial volume before z-scoring, and (iii) a modality-reweighted variant in which SBR received higher weight in the nigrostriatal pathway and volume contributed more strongly in microvascular pathways, guided by prior imaging literature.

For each specification, we recomputed MPIS and repeated MPIS–clinical association analyses. We quantified robustness by the proportion of MPIS–clinical associations that remained directionally consistent and FDR-significant across specifications, and the rank correlation between effect sizes across MPIS variants. Sensitivity results summarized in the Results demonstrate that the main pathway-level conclusions do not hinge on a single arbitrary choice of weighting or normalization.

## A.5 PATHWAY DEFINITIONS AND MULTIMODAL FEATURE MAPPING

### A.5.1 FEATURE CONSTRUCTION

Building on the QC'ed T1, DTI, and DaTSCAN data described in Section 2, we construct a unified set of regional imaging features and compress them into pathway-level integrity scores. At the subject level, we assemble a subject×feature matrix whose columns are ROI–modality measurements (T1 volumes, FA/MD from DTI, and striatal SBR from DaTSCAN), together with selected bilateral and asymmetry summaries. For downstream analyses and ablations, these features are organized into four nested views (V1–V4) that progressively add structural, dopaminergic, and diffusion information, as summarized in Table 2. To impose a biologically interpretable circuit structure, we further group ROIs into six PD-relevant pathway bins (nigrostriatal, frontostriatal–executive, cerebello–

Table 7: Functional pathway groupings of ROIs for pathway-anchored co-clustering analysis

| Pathway System | ROI Count | Key Structures (FastSurferCNN indices) | PD-Relevant Symptoms |
|---|---|---|---|
| **Motor Pathways** | | | |
| Nigrostriatal | 8 | Caudate (8,29), Putamen (9,30), Pallidum (10,31), Thalamus (7,28) | Bradykinesia, rigidity, tremor, motor slowing |
| Cortico-basal ganglia-thalamo-cortical | 12 | Precentral (62,94), Postcentral (60,92), Striatum (8,9,29,30), Thalamus (7,28), Cerebral WM (1,22) | Voluntary movement initiation, motor coordination |
| Corticospinal tract | 4 | Cerebral WM (1,22), Brainstem (13), Precentral (62,94) | Contralateral motor deficits, bradykinesia |
| Cerebellar motor | 10 | Cerebellar cortex (6,27), WM (5,26), Vermis (19–21), Deep nuclei (CerebNet 17–22) | Postural instability, gait, tremor modulation |
| **Cognitive Pathways** | | | |
| Frontostriatal executive | 16 | Dorsolateral prefrontal (42,65,74), Caudate (8,29), ACC (41,64,73), Rostral middle frontal (65) | Executive dysfunction, planning, working memory |
| Frontostriatal cognitive-motor | 6 | Caudate (8,29), Thalamus (7,28), Precentral (62,94) | Decision-making, motor planning integration |
| **Limbic Pathways** | | | |
| Mesolimbic reward | 4 | Accumbens (16,34), Ventral striatum components | Apathy, anhedonia, motivation deficits |
| Hippocampal memory | 8 | Hippocampus (14,32), Parahippocampal (54,86), Entorhinal (44,76) | Memory impairment, early cognitive decline |
| Amygdala emotional | 4 | Amygdala (15,33), Cingulate (41,48,61,73,80,93) | Anxiety, depression, emotional blunting |
| **Sensory & Association Pathways** | | | |
| Visual processing | 8 | Occipital cortex (43,49,51,75,81,83), Optic radiation (HypVINN 11,12) | Visual hallucinations, visuospatial deficits |
| Parietal attention | 4 | Inferior parietal (46,78), Superior parietal (67) | Impaired visuospatial attention |
| **Autonomic & Homeostatic Pathways** | | | |
| Hypothalamic autonomic | 12 | Anterior/middle/posterior hypothalamus (HypVINN 1–6), Preoptic (19,20) | Sleep disturbances, autonomic dysfunction, thermoregulation |
| Brainstem arousal | 3 | Brainstem (13), Periventricular gray (HypVINN 17,18,21,22) | REM sleep behavior disorder, sleep fragmentation |

ROIs grouped by established neurobiological circuits implicated in PD pathophysiology. Indices refer to FastSurferCNN labels unless otherwise noted. These pathway masks are used for pathway-anchored co-clustering to test whether discovered feature clusters align with mechanistic circuit boundaries.

thalamo–cortical, limbic/mesolimbic, microvascular-burden, and sensory/visuospatial), whose ROI and modality composition is specified in Table 9 and illustrated schematically in Figures 2 and 5.

### A.5.2 ROI-LEVEL FEATURE CONSTRUCTION AND PATHWAY BINNING

From the preprocessing pipeline we obtain regional features derived from T1-weighted MRI (volumes), DTI (FA and MD), and DaTSCAN SPECT (striatal uptake and specific binding ratios, SBR) for each patient. Using the DKT-based segmentation in T1-native space, we compute left and right hemisphere values for each ROI and modality and, where appropriate, bilateral means and asymmetry indices (Asym = (R–L)/(R+L)). These regional measurements instantiate the columns of the subject×feature matrix $X \in \mathbb{R}^{N \times D}$ introduced above, with each row representing one participant and each column one ROI–modality–hemisphere (or asymmetry) feature. Non-numeric metadata fields and derived "-std" columns are excluded. Features are standardized via a global $z$-transform, so that each column of $X$ has zero mean and unit variance before downstream analyses.

To operationalize a "pathway-centric" representation, we then map these ROI-level features into the predefined circuit bins summarized in Table 9. Consistent with the overview above, anatomically related ROIs are grouped into six PD-relevant pathways. For each pathway, we specify the set of cortical and subcortical ROIs assigned to the circuit and the modalities available for those ROIs. Figures 2 and 5 provide schematic illustrations of these circuits, while Table 9 makes explicit the mapping from the abstract notion of "pathway-anchored stratification" to the underlying ROI–modality features used in our analyses.

### A.6 SUPPLEMENTARY RESULTS

#### A.6.1 CLUSTER COMPOSITION AND COVARIATES

To assess potential confounding and to clarify the clinical meaning of the imaging-driven clusters, we first examined their composition with respect to diagnosis and standard demographic/technical covariates. Table 10 summarizes, for each global cluster, the number of participants and the distribution of baseline diagnosis (PD, SWEDD, control), age, sex, years of education, disease duration (for PD/SWEDD), levodopa-equivalent daily dose, and scanner field strength, as well as the marginal distributions of the primary clinical scales.

Table 8: Multimodal imaging features organized by neuroanatomical system and PD relevance

| System | Regions of Interest | Modality | Metrics |
|---|---|---|---|
| *Nigrostriatal* *(primary motor)* | Caudate, Putamen, Accumbens (bilateral) | DaTSCAN | SBR, mean, std |
| | Caudate, Putamen, Accumbens (bilateral) | T1 MRI | Volume |
| | Caudate, Putamen (bilateral) | FWE-DTI | FA$_T$, MD$_T$, FW (mean, std) |
| *Thalamo-cortical* *(integration hub)* | Thalamus (bilateral) | T1 MRI | Volume |
| | Thalamus (bilateral) | FWE-DTI | FA$_T$, MD$_T$, FW (mean, std) |
| *Limbic* *(cognition/emotion)* | Hippocampus, Amygdala, Accumbens (bilateral) | T1 MRI | Volume |
| | Hippocampus, Amygdala (bilateral); parahippocampal ctx (bilateral) | FWE-DTI | FA$_T$, MD$_T$, FW (mean, std) |
| *Cortical* *(executive/cognitive)* | Orbitofrontal, rostral middle frontal, superior temporal, insula, cingulate (bilateral) | T1 MRI | Volume |
| | Insula, superior temporal, rostral middle frontal, parahippocampal (bilateral) | FWE-DTI | MD$_T$, FW (mean, std) |
| *White matter tracts* *(connectivity)* | Cerebral WM (bilateral), corpus callosum | FWE-DTI | FA$_T$, MD$_T$, FW (mean, std) |
| | Corpus callosum | — | Cognitive decline marker |
| *Brainstem &* *cerebellum* | Brainstem | T1 MRI | Volume |
| | Brainstem, cerebellum WM, cerebellum cortex | FWE-DTI | FA$_T$, MD$_T$, FW (mean, std) |

Table 9: Definition of pathway bins used for MPIS computation. Modalities refer to the measurement types pulled per ROI (Volume, FA, MD, SBR). Feature counts $|\mathcal{F}_p|$ were computed from `metrics_final.csv` with the current pathway bins.

| Pathway | Key regions (examples) | Modalities included | $|\mathcal{F}_p|$ |
|---|---|---|---|
| Nigrostriatal (motor) | Substantia nigra[*], putamen, caudate, globus pallidus, thalamus, primary motor/premotor cortex | Volume + FA + MD + SBR (4 modalities) | 88 |
| Frontostriatal–executive | Dorsolateral PFC, anterior cingulate, insula, caudate, putamen, thalamus | Volume + FA + MD + SBR (striatal nuclei) | 128 |
| Cerebello–thalamo–cortical (CTC) | Cerebellar hemispheres/WM, dentate pathway, thalamus, SMA/motor cortex | Volume + FA + MD (3 modalities) | 20 |
| Limbic / mesolimbic | Hippocampus, amygdala, nucleus accumbens, orbitofrontal cortex, anterior/posterior cingulate | Volume + FA + MD + SBR (ventral striatum) | 80 |
| Microvascular-burden | Periventricular/deep WM lesions (WM-hypointensities) | Volume + FA + MD (3 modalities) | 4 |
| Sensory / visuospatial | Occipital cortex, posterior parietal cortex, temporal–parietal junction, auditory cortex | Volume + FA + MD (3 modalities) | 112 |

By construction, the SRVCC clustering algorithm did not use any clinical or covariate information; clusters are defined solely in the multimodal imaging space. Consistent with this design, the covariate summaries in Table 10 show no extreme demographic or acquisition imbalances that would trivially explain the observed imaging structure (for example, no cluster is composed exclusively of a single diagnostic group or a single scanner field strength). Instead, PD and SWEDD participants are represented across multiple clusters, and age, sex, and education show overlapping ranges, suggesting that the global clusters primarily capture latent imaging phenotypes rather than obvious sampling artefacts.

### A.6.2 PATHWAY-SPECIFIC PROFILES

- Nigrostriatal thalamo–cortical (motor) pathway. The nigrostriatal pathway showed the clearest MPIS–motor coupling (Table 4): lower MPIS was associated with higher motor severity (MDS-UPDRS III: $\rho \approx -0.201$, $q \approx 6.8 \times 10^{-4}$), with negligible associations with MoCA and QUIP_SUM. MPIS separated strongly across pathway clusters ($H_{\text{MPIS}} \approx 167.15$, $\eta^2 \approx 0.587$), indicating robust within-pathway stratification. Detailed per-outcome cluster tests and feature-driver analyses are reported in Supplementary Results.

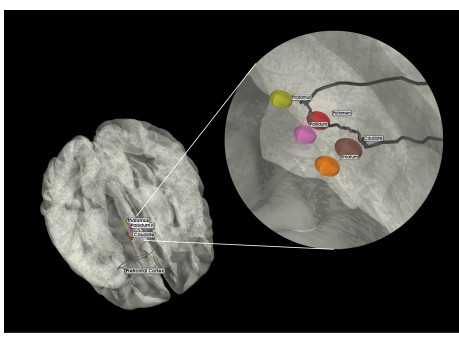 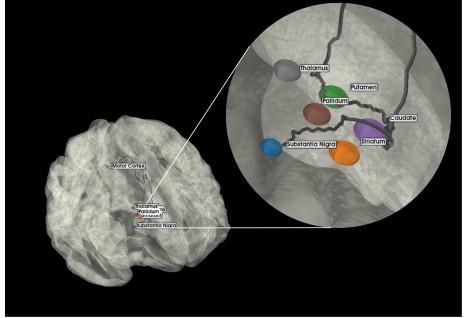

(a) Frontostriatal pathway (cognitive/executive/attention)

(b) Nigrostriatal pathway (motor basal ganglia–thalamo–cortical)

Figure 5: Pathway schematics of (A) frontostriatal (cognitive/executive/attention) and (B) nigrostriatal (motor basal ganglia–thalamo–cortical).

- Sensory / visuospatial pathway. This pathway showed the strongest MPIS–cognition association (Table 4): higher MPIS was associated with better global cognition (MoCA: $\rho \approx 0.163$, $q \approx 0.0071$), with weaker, non-significant relationships to MDS-UPDRS III and QUIP_SUM. MPIS separated strongly across pathway clusters ($H_{\mathrm{MPIS}} \approx 170.68$, $\eta^2 \approx 0.629$). Full per-outcome cluster tests and feature-driver analyses are provided in Supplementary Results.

- Frontostriatal cognitive (executive/attention) pathway. Frontostriatal MPIS showed a robust negative association with motor severity (MDS-UPDRS III: $\rho \approx -0.191$, $q \approx 0.0014$) and strong MPIS separation across clusters ($H_{\mathrm{MPIS}} \approx 121.50$, $\eta^2 \approx 0.432$), with weaker global cognitive and QUIP_SUM associations (Table 4). Full figures and tables are reported in Supplementary Results.

- Limbic / mesolimbic pathway. Limbic MPIS demonstrated strong imaging stratification across clusters ($H_{\mathrm{MPIS}} \approx 165.25$, $\eta^2 \approx 0.58$), with a modest positive association with cognition (MoCA: $\rho \approx 0.119$, $q \approx 0.045$) and weaker motor and behavioral coupling (Table 4). Full figures and tables are provided in Supplementary Results.

- Microvascular burden (gait/cognition modifier) pathway. Microvascular MPIS stratified imaging profiles across clusters ($H_{\mathrm{MPIS}} \approx 127.11$, $\eta^2 \approx 0.445$) but showed near-zero correlations with MDS-UPDRS III, MoCA, and QUIP_SUM (all $q \gg 0.1$; Table 4), consistent with a modifier interpretation and motivating more targeted gait and dysexecutive endpoints. Full figures and tables are reported in Supplementary Results.

- Cerebello–thalamo–cortical (balance) pathway. CTC/balance analyses in this run were imaging-only (MPIS–clinical correlations not computed); however, clusters separated strongly, with dominant contributions from cerebellar morphometry and diffusion metrics. Full figures and the top-feature summary are provided in Supplementary Results.

## ABLATION ACROSS FEATURE VIEWS

We evaluated robustness of the co-clustering and MPIS construction across four feature views (2) ranging from clinical-only to the full multimodal set.

With clinical scores alone (V1), the resulting clusters were coarser, less stable across resamples, and offered limited anatomical specificity: cluster-defining differences primarily reflected overall severity gradients rather than distinct circuit-level profiles. Adding structural volumes and DaT-SBR (V2) sharpened separation, particularly in striatal and limbic systems, and improved alignment with motor burden, but still yielded weaker microstructural differentiation. Incorporating diffusion metrics (V3) produced further gains in within-cluster homogeneity, although uncorrected DTI introduced modest redundancy with T1 volumes. The full multimodal configuration with free-water–corrected diffusion (V4; primary analysis) provided the most stable clusters and clearest pathway-level interpretations, with MPIS patterns and top separating features consistent across bootstrap runs. Together, these ab-

Table 10: Cluster-wise composition and key covariates derived from the FA/MD run. Counts are total (PD / SWEDD); summary values are mean $\pm$ SD.

|  | Cluster 1 | Cluster 2 | Cluster 3 | Cluster 4 | Cluster 5 |
|---|---|---|---|---|---|
| $n$ (PD / SWEDD) | 32 (23/9) | 28 (24/4) | 51 (44/7) | 44 (36/8) | 12 (10/2) |
| Age (years) | $63.0 \pm 8.6$ | $59.8 \pm 7.0$ | $65.6 \pm 7.7$ | $54.4 \pm 7.3$ | $71.6 \pm 5.5$ |
| Female (%) | 37.5 | 25.0 | 39.2 | 56.8 | 16.7 |
| MDS-UPDRS III (NP3TOT) | $19.7 \pm 9.9$ | $21.4 \pm 9.0$ | $21.2 \pm 9.2$ | $16.9 \pm 9.5$ | $20.7 \pm 9.1$ |
| MoCA (MCATOT) | $27.6 \pm 1.8$ | $27.9 \pm 1.8$ | $27.3 \pm 2.4$ | $28.2 \pm 1.6$ | $27.5 \pm 1.1$ |
| QUIP_SUM | $4.1 \pm 1.9$ | $5.1 \pm 1.1$ | $4.9 \pm 1.7$ | $4.3 \pm 1.8$ | $4.5 \pm 1.7$ |

lations indicate that our pathway-specific conclusions are not driven by a single modality, but reflect convergent signal from structure, dopaminergic binding, and tissue microstructure.

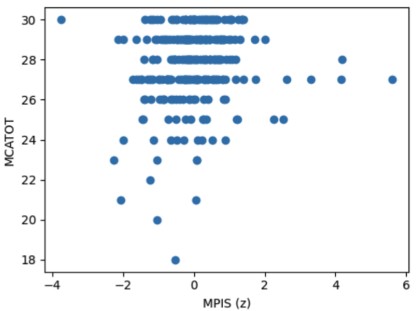 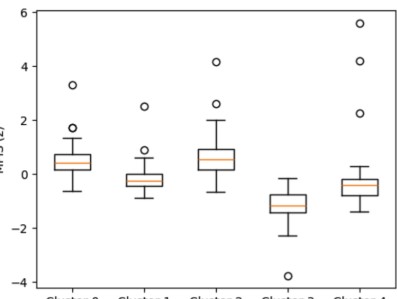

Figure 6: Sensory/visual/visuospatial MPIS associations. Left: MPIS vs MoCA (Spearman $\rho \approx 0.163$, $q \approx 0.0071$). Right: MPIS separation across clusters (Kruskal–Wallis $H \approx 170.68$, $\eta^2 \approx 0.629$).

