# OpenReview forum: "Brain pathway anchored multimodal generative representations for patient-specific predictions of Parkinson’s disease"
_ICLR.cc/2026/Workshop/LMRL — Submitted to ICLR 2026 Workshop LMRL_

### Official Review · Reviewer_Gb5H · 2026-02-23
**Well-executed application study, but limited ML contribution**

**Rating:** 3
**Confidence:** 4

**Review:**

This paper presents a pathway-anchored analysis of multimodal neuroimaging data (structural MRI, diffusion MRI, DAT-SPECT) in Parkinson's disease using the PPMI cohort. The authors apply SRVCC clustering to identify patient subtypes and define Multimodal Pathway Integrity Scores (MPIS) that aggregate imaging features within predefined neuroanatomical circuits. The work is technically competent with rigorous quality control, stability analyses, and appropriate statistical methods. The pathway-specific MPIS values show biologically coherent correlations with clinical measures (motor severity, cognition), and the results align well with established PD pathophysiology. However, the MPIS are hand-crafted weighted sums of z-scored features (Equation 1), and SRVCC is directly imported without modification. The paper offers limited methodological innovation and the insights are not transferable to other representation learning problems.

---

### Meta-Review · Area_Chair_eiQb · 2026-02-25

**Recommendation:** Reject
**Confidence:** 3

**Metareview:**

Recommend rejection

---

### Decision · Program_Chairs · 2026-03-02

**Decision:**

Reject

**Comment:**

Please see the meta-review.